# High-Resolution Displacement Sensor Based on a Chirped Fabry–Pérot Interferometer Inscribed on a Tapered Microfiber

**Zhong Lu, Yuan Cao \*, Guangying Wang, Yang Ran, Xinhuan Feng and Bai-Ou Guan**

Guangdong Provincial Key Laboratory of Optical Fiber Sensing and Communications, Institute of Photonics Technology, Jinan University, Guangzhou 510632, China; xylylu@163.com (Z.L.); guangyingwang@stu2017.jnu.edu.cn (G.W.); tranyang@jnu.edu.cn (Y.R.); eexhfeng@gmail.com (X.F.); tguanbo@jnu.edu.cn (B.-O.G.)

**\*** Correspondence: qianshuizhong@126.com; Tel.: +86-155-2107-4168

**Abstract:** In this study, a high-resolution optical fiber Fabry–Pérot (FP) interferometer displacement sensor with chirped spectral characteristics based on a tapered microfiber is theoretically discussed and experimentally implemented. Instead of inscribing two fiber Bragg gratings (FBGs) on the symmetric position of the microfiber, we continuously inscribed one long FBG along the microfiber region to reduce the cavity length. The bandwidth of the interferometer is over 35 nm, and its displacement sensitivity is as high as 36.5 nm/mm at the tension state.

**Keywords:** optical fiber displacement sensor; Fabry–Pérot Interferometer; tapered microfiber

## 1. Introduction

Displacement is an important structural mechanical parameter that is used for identifying damage. The optical interferometer type of displacement sensor has the advantage of high sensitivity, small size, and immunity to electromagnetic interference. The interferometer type displacement sensor usually has multiple peaks and deeps. When demodulating the wavelength shift of a sensor with a uniform free spectra range (FSR), it is very difficult to mark one of the (notch) wavelengths as the sensing wavelength [1–3], because when the sensing wavelength shift is more than one FSR, it is hard to trace the chosen sensing wavelength. However, when the FSR changes monotonically, the tracing of the chosen sensing wavelength becomes very easy [4–6]. Moreover, we can conduct a cross-correlation process to the chirped spectra to achieve a single peak. Marking one peak in the spectra is easier compared with the marking of multiple peaks, especially in the high-speed time domain-sensing demodulation system [7,8]. The spectral chirped optical fiber sensor can be also used in microwave photonics [9]. For example, in the field of microwave photonics radar systems, the chirped microwave signals can be generated with optical spectral-shaping techniques and frequency-to-time mapping techniques.

The conventional structures to achieve the chirped shape of optical spectra are combining double arm Mach–Zehnder interferometers [7] or Sagnac fiber loop interferometers [8] with the chirped fiber Bragg gratings (CFBGs). However, the long fiber in the interferometer structure is very sensitive to environmental disturbance and also difficult to integrate due to its large size. Optical fiber Fabry–Pérot (F–P) interferometers have been extensively investigated due to their high sensitivity, low cost, and compact size, but the FSR of an F–P interferometer is usually uniformed. The optical fiber taper is a key component that has a number of applications in the fiber optic area, such as optical fiber couplers, filters, and sensors. The optical fiber taper has the advantages of the ease of fabrication, stability from environmental changes, and compactness. Such applications mentioned above have

attracted huge research attention in the past few years [10–13]. Moreover, the optical fiber taper can also achieve CFBG due to the broadband characteristics of microfiber Bragg grating (mFBG) [14–17]. Usually the microfiber F–P interferometer is achieved by inscribing two FBGs at the two taper regions of the microfiber. However, the monotonically decreasing FSR is hard to achieve, because the wavelength ranges of the spectra of the two FBGs are usually mismatched [4–6].

In this paper, we propose and experimentally demonstrate a spectral chirped Fabry–Pérot (F–P) interferometer displacement sensor based on a tapered microfiber. We continuously inscribed a long uniform FBG onto the microfiber by scanning the focus spot of a 193-nm ArF excimer laser with a phase mask (1072 nm). The available bandwidth of the chirped F–P interferometer is over 20 nm. We marked the notch wavelength of the reflective spectra under different tensions to show the good tunability of the F–P interferometer chirp rate. The cross-correlation analyzing for the reflective spectra is also given to enhance the sensing demodulation efficiency. The displacement sensing curves created by marking the first notch wavelength shifting and by a cross-correlation analyzing method are both given in the experimental results. The proposed chirped F–P interferometer has potential applications in the fields of high-speed stress measurement and microwave signal generation.

## 2. Operation Principle

The spectral characteristics of an interferometer (including the free spectra range and the chirp rate) are strongly dependent on the wavelength-to-phase relationship at the interference port. The key point of the fabrication of the chirped F–P interferometer is to realize the highly nonlinearly changing wavelength-to-phase relationship. Thus, the optical path difference should depend on the wavelength. The optical microfiber can perfectly meet this condition. When the microfiber diameter reduces to only several micrometers, the effective refractive index $n_{eff}$ strongly depends on the microfiber diameter [16]. The microfiber diameter-to-$n_{eff}$ relationship calculated by the COMSOL software is shown in Figure 1a.

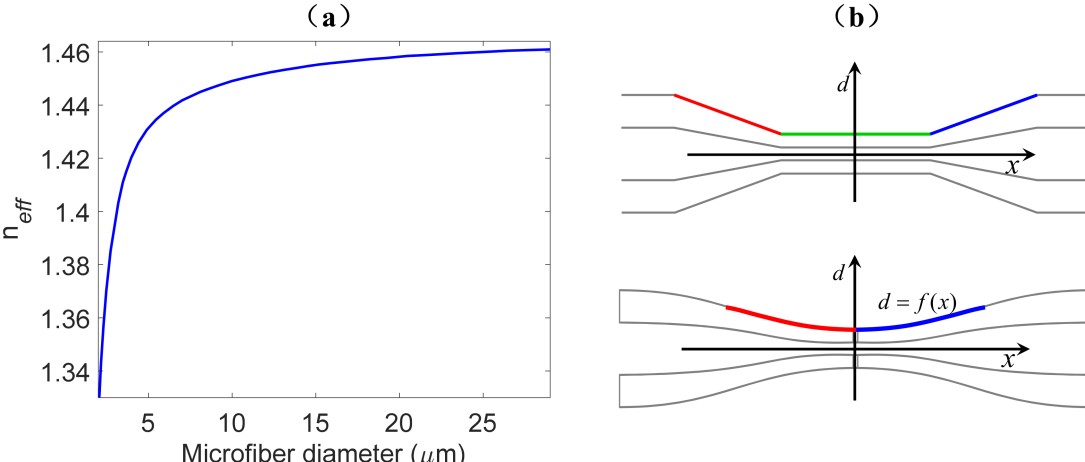

**Figure 1.** (**a**)The microfiber diameter-to-$n_{eff}$ relationship curve calculated by the COMSOL software; (**b**) the traditional tapered fiber analysis model (up) with 3 regions and the proposed tapered fiber analysis model (down) with 2 slowly changing regions (*d* is the diameter of the tapered microfiber and x is the position along the fiber).

As the diameter decreases, the effective refractive index of the microfiber $n_{eff}$ also declines and drops dramatically with the decreasing of the fiber diameter when the diameter is less than 7 nm [16]. Figure 2 shows the fabrication and sensing process of the proposed chirped F–P interferometer. A standard multimode fiber (62.5 μm/125 μm) is drawn and tapered by butane flame brushing. The horizontal cross section of the microfiber conforms to a certain outline shape. In traditional tapered fiber analysis models, the microfiber is divided into two tapers with linearly changing diameters and a uniform region to simplify the simulation process, as shown in Figure 1b. However, the existence of a

uniform region does not satisfy the real situation, especially for the analyzing of interferometer-type optical sensors. Thus, we consider the microfiber to be two slowly changing regions with a specific outlined shape. In the grating fabrication step, by adding periodical refractive index modulation of the microfiber with a phase mask and excimer laser, a broadband fiber Bragg grating can be achieved. The detailed analyses are shown as follows.

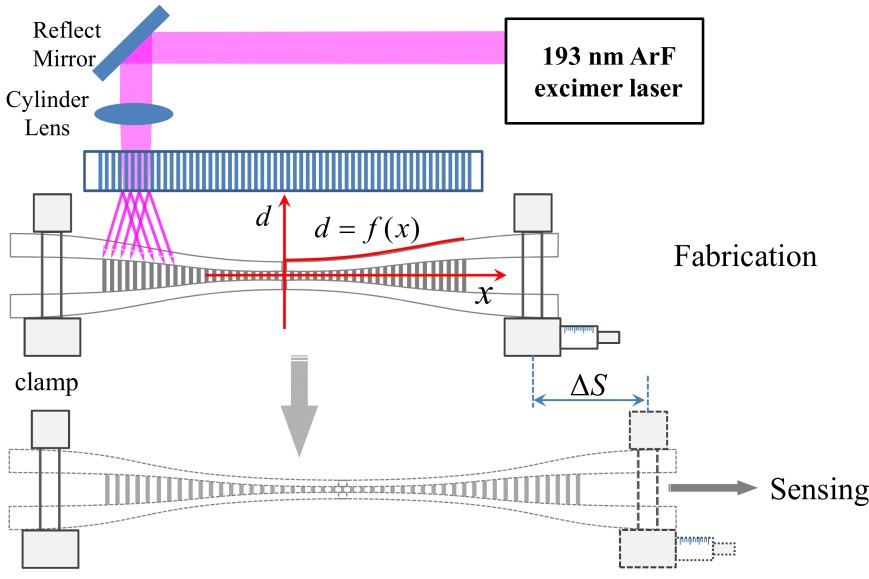

**Figure 2.** The fabrication and sensing process of the proposed chirped Fabry–Pérot (F–P) interferometer.

Here, we consider the two taper regions to be symmetrical and the origin of the coordinates is at the position of the minimum radius. As the red curve shown in Figure 2, the diameter $d$ of the tapered microfiber is the function of the position along the fiber $x$ that can be expressed as follows:

$$d = f(x). \tag{1}$$

According to the coupled-mode theory, the reflected optical wavelength satisfies the phase matching condition when the Bragg grating is formed. Then, the wavelength at position x is also related to x, which is expressed as follows:

$$\lambda(x) = n_{eff}(f(x)) \cdot \Lambda \tag{2}$$

where $\wedge$ is the grating period of the phase mask. Due to the monotonicity of $f(x)$ and $n_{eff}$, (2) can also be expressed as follows:

$$x = f^{-1}(n_{eff}^{-1}(\frac{\lambda}{\Lambda})) \tag{3}$$

where $f^{-1}(x)$ and $n^{-1}_{eff}(d)$ are the inverse functions of $f(x)$ and $n_{eff}(d)$ (the curve in Figure 1a), respectively. Then, the phase difference of λ when the light propagates back and forth in the cavity can expressed as follows:

$$\Delta\varphi(\lambda) = 4x/\lambda \cdot 2\pi + \pi = f^{-1}(n_{eff}^{-1}(\frac{\lambda}{\Lambda}))/\lambda \cdot 8\pi + \pi. \tag{4}$$

The extra $\pi$ in Equation (4) is caused by the half-wave loss. The phase change between the different wavelengths will keep changing slowly and monotonously. The amplitude reflectance can be expressed as follows:

$$R(\lambda) = r_1(\lambda) + \frac{(1 - r_1^2(\lambda))^2 \cdot r_2(\lambda) \cdot e^{i\Delta\varphi(\lambda)}}{1 - r_1(\lambda) \cdot r_2(\lambda) \cdot e^{i\Delta\varphi(\lambda)}} \tag{5}$$

where $r_1(\lambda)$ and $r_2(\lambda)$ are the reflectance of the microfiber Bragg gratings inscribed on the two tapers. If we ignore the refractive index modulation depth difference induced by the UV absorption efficiency change, $r_1(\lambda)$ and $r_2(\lambda)$ can be approximated as equal. It should to be noted that Equation (5) is under the condition that the wavelength range of the two FBGs are inscribed at the exact, symmetrical position, which is very difficult to achieve. Otherwise, there will be no interference at the mismatched wavelength area at both the short wavelength region and the long wavelength region, as shown in Figure 3a.

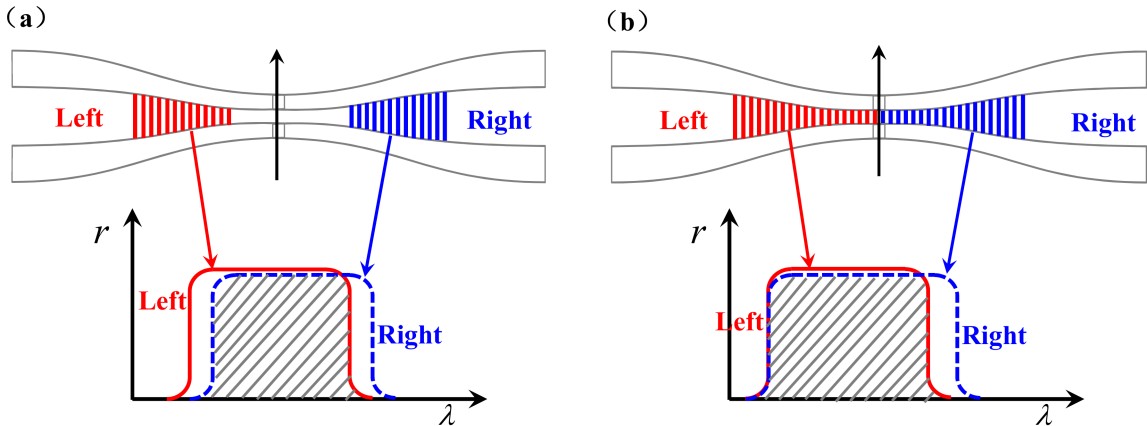

**Figure 3.** (**a**) The structure of the two separated fiber Bragg gratings (FBGs)-based F–P interferometer and the reflected spectra of the FBGs; (**b**) The structure of the FBG-based F–P interferometer and the reflected spectra of the FBG.

We can see that the two FBGs have the same length, but the left FBG (the red grating) is close to the zero point, while the right FBG is far from the zero point (the blue grating). Then, the short wavelength of the left FBG spectra is smaller than that of the right FBG spectra, and the long wavelength of the left FBG spectra is smaller than that of the right FBG spectra. The overlapping region (the dashed area in the spectra) is where the interference occurs, and there is no interference region in the short wavelength range. Unlike the traditional fabrication process of the F–P interferometer with two discrete FBGs, one long FBG inscribed across the two taper regions is used in our experiment, as shown in Figure 3b. We can see that the short wavelengths of the two FBGs are the same. At the zero point, the length of the cavity is zero, which means that the phase difference at the short wavelength is $\pi$ (considering the half-wave loss). As the position moves away from the zero point, the phase difference increases with the wavelength, and the overlapping region (the dashed area) is the interference region. There will be extra spectra range for the right FBG, which can be easily filtered out by additional optical filters. When the tension is given to the displacement sensor, the profile shape function $f(x)$ will change, and the reflective spectra will also change. Moreover, a microfiber can be more easily stretched than the conventional optical fiber due to its low stiffness, which is beneficial in the application of displacement sensing.

## 3. Experimental Results

In our experiment, the fiber taper was fabricated by a flame-heated taper-drawing technique. The minimum diameter of the fiber taper was about 3.71 μm, and the length of the whole taper region was about 1.9 cm. In the chirped F–P interferometer fabrication process, we used a uniformed phase mask with $\wedge$ equal to 1072 nm to inscribe the Bragg grating. The chirped grating was mounted onto two fiber clamps, and the distance between the two clamps was 19.06 mm. In the sensing experiment, one clamp was fixed while the other moved along the fiber axis. Figure 4 shows the measured reflected spectra of the F–P interferometer with different tension.

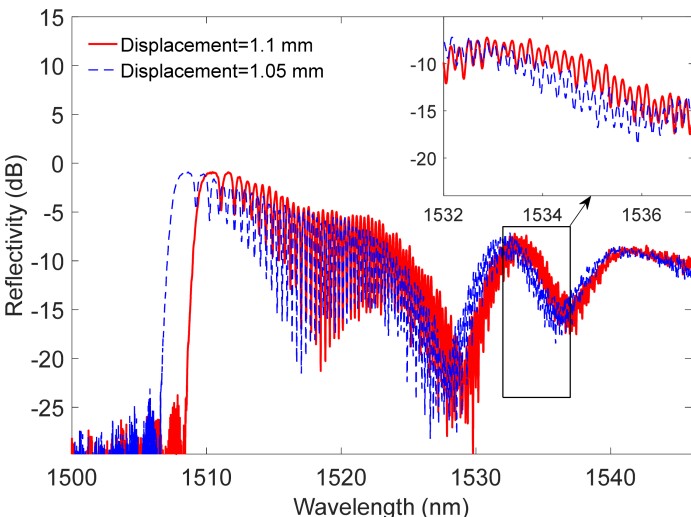

**Figure 4.** The F–P interferometer experimentally reflected spectra under displacements of 1.05 mm (blue dot line) and 1.1 mm (red line).

The full bandwidth of the F–P interferometer spectra was more than 35 nm, but there were two valleys in the long wavelength region (at 1527.8 nm and 1536.3 nm for spectra under displacement of 1.05 mm). These valleys may have been caused by the steep decline of the reflectance. Although the scanning speed of the ArF excimer laser beam was constant, the slope of outline shape function changed with the position along the fiber, which means that the FBG at different positions received different laser powers. When the slope of the outlined shape function reached its maximum value, the power absorption efficiency reached its minimum value. The reflective spectra at the wavelengths before the valleys are relative flat and can be used for chirped optical shaping. Thus, the available bandwidth of the F–P interferometer was over than 20 nm. When the displacement was increased by 0.05 mm, the reflected spectra were changed from the blue curve to the red curve, and the wavelength shift marked by the first notch was 1.86 nm. To investigate the chirp characteristics of the sensor, we marked the first 60 notches of the spectra when the displacement was between 0.9 mm and 1.1 mm, as shown in Figure 5.

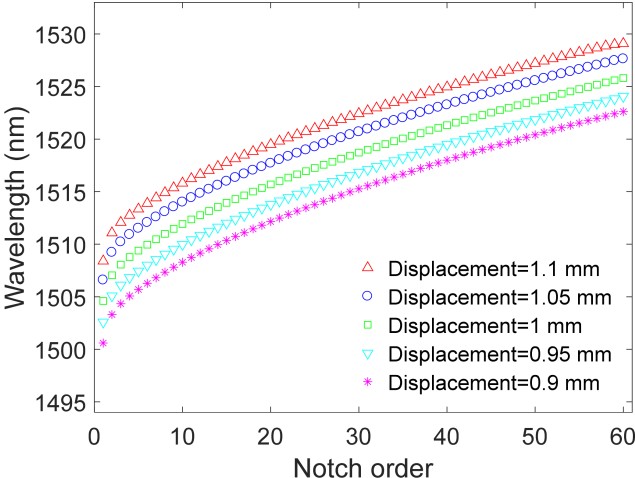

**Figure 5.** The first 60 notches of the F–P interferometer-reflected spectra under different displacements.

It can be seen that when the displacement introduced to the F–P interferometer increases, the notches in the reflected spectra shift to the long wavelength region, and the differential of the notches' wavelength also change slightly. This is because the stiffness along the fiber is not constant,

and the FBG period changes more slowly when the fiber diameter is larger. The displacement sensing can be achieved by marking the wavelength of the first notch, but the marking step usually needs manual work due to the existence of multiple wavelength spectra. By adding a cross-correlation process to the reflected spectra, only one peak needs to be marked. The principle of the cross-correlation process is discussed in [8]. The wavelength of the notches in the reflected spectra means that the phase difference is an odd multiple of π. Thus, we can build a reference function based on this wavelength to the phase and do the cross-correlation process to the reflected spectra. The cross-correlation result under different tensions is shown in Figure 6.

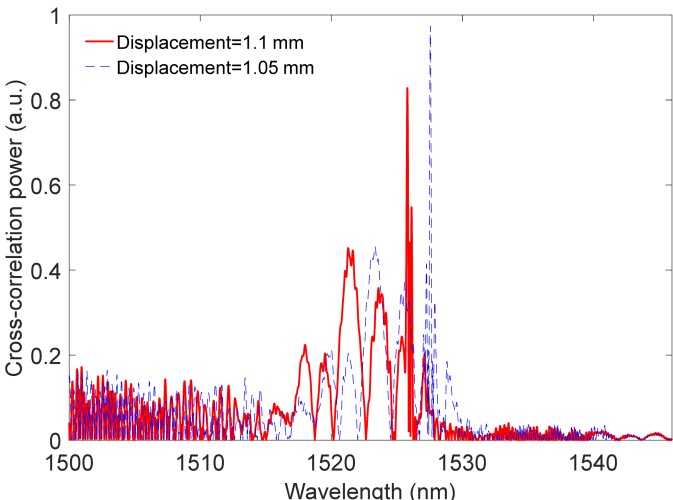

**Figure 6.** The cross-correlation curves under a displacement of 1.05 mm (blue dot line) and 1.1 mm (red line).

The cross-correlation curve has one narrow peak and other side lobes. The separation of the peaks between the different displacements is clear enough to recognize the displacement changes below. The sensing curve that was achieved by marking the first notch wavelength and the cross-correlation peak wavelength is shown in in Figure 7.

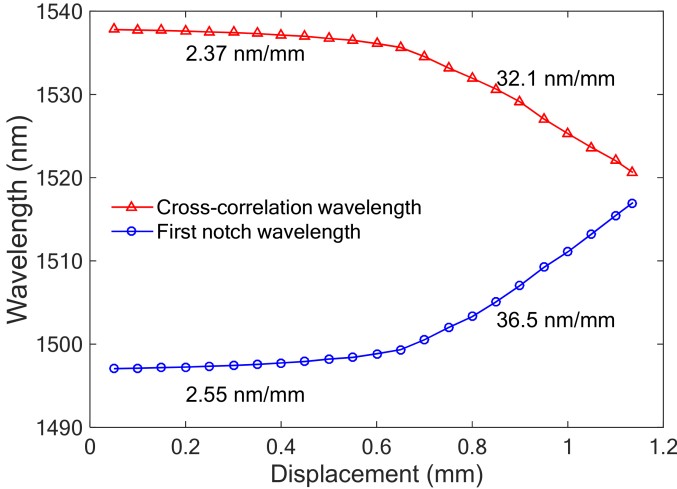

**Figure 7.** The displacement sensing curve of the notch wavelength shifting (blue line) and cross-correlation peak shifting (red line).

It can be seen that the first notch wavelength and the cross-correlation peak wavelength change with the displacement in opposite directions. The shifting curves of the two methods do not change

linearly with the displacement. The displacement sensitivity is low at the loose state of the microfiber and high at the tension state. For the notch wavelength marking method, the displacement sensitivity was 2.55 nm/mm at the loose state and 36.5 nm/mm at the tension state. For the cross-correlation method, the displacement sensitivity was 2.37 nm/mm at the loose state and 32.1 nm/mm at the tension state. There was a slight sensitivity difference between the two methods. However, the demodulation efficiency of the cross-correlation method is much higher than the wavelength marking method. The cross-correlation steps also suggest that the Fabry–Pérot interferometer has potential applications in the field of microwave signal generation. The measurement sensitivity of both the methods mentioned here is higher than the displacement sensitivity of 22.8 nm/mm mentioned in [3], and the bandwidth of the proposed F–P (0.15) is far less than that of [3] (20 nm). The sensing resolution of the proposed F–P is 0.004 mm, calculating from the ratio of the 3-dB bandwidth to the sensitivity. The displacement sensing can be converted into strain sensing after calculation, the strain sensitivity comparison between the proposed F–P interferometer and other interferometer-type sensors are shown in Table 1.

**Table 1.** Comparisons between the references based on their interferometer-type sensors and the proposed sensor.

| Sensor | Sensitivity | 3-dB Bandwidth | Ratio of Bandwidth to Sensitivity |
|---|---|---|---|
| The proposed F–P | 6.4 pm/$\mu\varepsilon$ | 0.15 nm | 23.4 $\mu\varepsilon$ |
| M-Z [1] | 1.19 pm/$\mu\varepsilon$ | 4.3 nm | 3613 $\mu\varepsilon$ |
| SMS structure [2] | 42.5 pm/$\mu\varepsilon$ | 7.3 nm | 171.8 $\mu\varepsilon$ |

We can see that the sensitivity of the proposed F–P interferometer is higher than the Mach-Zehnder (M–Z) structure, while the 3-dB bandwidth is much narrower. The sensitivity of the proposed F–P interferometer is less than that of the single mode-multimode-single mode (SMS) structure, but the 3-dB bandwidth of the proposed F–P interferometer is much less than that of the SMS structure. The ratio of the bandwidth to the sensitivity is a very important parameter, which is often used to represent the sensing resolution. The sensing resolution of the proposed F–P interferometer is higher than the two common interferometer-type sensors. We also measured the temperature sensitivity, as shown in Figure 8.

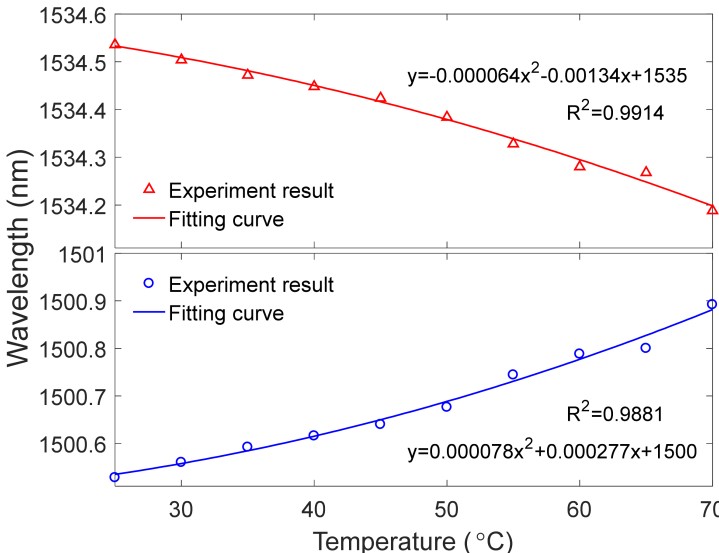

**Figure 8.** The temperature sensing curve of the notch wavelength shifting (blue line) and cross-correlation peak shifting (red line).

We can see from the temperature curve that for both the cross-correlation method and the notch wavelength shift monitoring method, the sensitivity is changing with the temperature as a quadratic

function. The temperature sensitivity was about 0.04 nm/°C at room temperature (25 °C), which was even lower than the optical spectra analyzer (OSA) resolution. Thus, the temperature variations to the displacement sensing can be ignored.

## 4. Conclusions

A chirped Fabry–Pérot interferometer displacement sensor with long FBG inscribed on the microfiber was experimentally demonstrated and theoretically analyzed. The propose F–P interferometer can be used in a high-speed displacement sensing due to its continuously changing chirp rate. The displacement sensitivity was as high as 36.5 nm/mm at the tension state of the microfiber, and the temperature sensitivity was very low. Cross-correlation analyses were introduced to enhance the sensing demodulation efficiency. The proposed Fabry–Pérot interferometer has potential applications in the field of microwave signal generation.

**Author Contributions:** Original draft preparation, investigation, and validation, Z.L.; review and editing of the manuscript, conceptualization, and methodology, Y.C.; validation and formal analysis, G.W.; methodology, Y.R.; and project administration and supervision, X.F. and B.-O.G.

**Funding:** This research was funded by the National Natural Science Foundation of China (NSFC), (No. 61701193, No. 61771221 and No. 61860206002).

**Conflicts of Interest:** The authors declare no conflicts of interest.

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
