# Peer review of "High-Resolution Displacement Sensor Based on a Chirped Fabry–Pérot Interferometer Inscribed on a Tapered Microfiber"

_applsci, doi:10.3390/app9030403_

Round 1
Reviewer 1 Report
The authors report on a displacement sensor based on a FBG and a microfiber. The results are interesting but the manuscript lacks of important information. Therefore, I believe the authors should improve on their paper before the journal can consider its publication. In the following, I provide a set of suggestions that may help the authors to improve the paper quality.
1) Please clarify the method used for obtaining Fig. 1 data (equation, numerical data…).
2) In some parts of the text, the authors refer to “wavelength” as “wave”. For example, they say “in the long wave region (at 1527.8nm…)”. Please change these terms to wavelength in order to avoid confusion.
3) Please add the fiber taper characteristics: diameter, length, etc.
4) Could the authors develop on how the sensor performance could be further improved? Are the setup parameters the optimized ones?
5) The authors say that they have a high resolution sensor, but they do not estimated it. Please provide this information.
6) Does the authors expect any cross-sensitivity due to temperature variations? Could the authors estimate that?
7) Is it possible, by using the notch wavelength shift and the cross-correlation information to probe two parameters simultaneously (say displacement and temperature)?
8) Comparison with other sensors results is missing.
Author Response
We thank the reviewer for the comments and recommendations.
1. Please clarify the method used for obtaining Fig. 1 data (equation, numerical data…).
The data in figure 1 is the simulation result using the COMSOL software. The data of figure 1(a) is also supported by reference [12]. We have added the corresponding description in line :
When the microfiber diameter reduces to only several micrometers, the effective refractive index neff strongly depends on the microfiber diameter [12]. The microfiber diameter to neff relation calculated by the COMSOL software is shown in figure 1 (a).
2. In some parts of the text, the authors refer to “wavelength” as “wave”. For example, they say “in the long wave region (at 1527.8nm…)”. Please change these terms to wavelength in order to avoid confusion.
Thank the reviewer for the careful review, we have corrected this spelling mistakes from line No. 105 to 111:
We can see that when the two FBGs have the same length, but the left FBG (the red grating) is close to the zero point, while the right FBG is far from the zero point (the blue grating), then the short wavelength of the left FBG spectra is smaller than that of the right FBG spectra, and the long wavelength of the left FBG spectra is smaller than that of the right FBG spectra. The overlapping region (the dashed area in the spectra) is where the interference occurs, and we can see that there is no interference region in the short wavelength range.
And in line No. 132:
The full bandwidth of the F-P interferometer spectra is more than 35 nm, but there are two deeps in the long wavelength region (at 1527.8 nm and 1536.3 nm for spectra under displacement of 1.05 mm).
3. Please add the fiber taper characteristics: diameter, length, etc.
The exact diameter cannot be measured because the process measured with the microscope may destroy the structure of the sensor. Thus we inversely calculate the minimum diameter by the diameter to neff relation curve in Fig. 1(a). The minimum diameter of the microfiber is 3.71μm. The length of the microfiber is 1.9 cm. We have added the fiber taper characteristics from line No. 121 to 123:
In our experiment, the fiber taper is fabricated by flame-heated taper-drawing technique, and the minimum diameter of fiber taper is about 3.71 μm and the length of the whole taper region is about 1.9 cm.
4. Could the authors develop on how the sensor performance could be further improved? Are the setup parameters the optimized ones?
The flatness of the spectra we obtained is still relatively poor. The main reason is that the inscribing process is uniform, and the relationship between the position of the microfiber and the wavelength is nonlinear, so the power obtained for each wavelength is different. The flatness of the spectra can be improved by inscribing with variable speed, so that the peak of the cross-correlation curve could be narrower. The optimization of the scanning speed is very difficult. We have established a theoretical model on the absorption efficiency, the experimental works are still in preparation.
5. The authors say that they have a high resolution sensor, but they do not estimated it. Please provide this information.
We have added the discussions about the displacement resolution from line No.177 to 181:
The measurement sensitivity of both methods mentioned here is higher than the displacement sensitivity of 22.8 nm/mm mentioned in [3], and the bandwidth of the proposed F-P (0.15) is far less than that of [3] (20 nm). The sensing resolution of the proposed F-P is 0.004 mm calculating from the ratio of the 3dB bandwidth to sensitivity.
6. Does the authors expect any cross-sensitivity due to temperature variations? Could the authors estimate that?
We added a temperature response experiment and the results show that the spectra drifts very slow with the temperature changing. The temperature sensitivity is about 0.04 nm/℃ at room temperature, which is lower than the OSA resolution. The corresponding description is from line No.193 to 201:
We also measured the temperature sensitivity as shown in figure 8.
Figure 8. The temperature sensing curve of the notch wavelength shifting (blue line) and cross-correlation peak shifting (red line).
We can see from the temperature curve that for both the cross-correlation method and the notch wavelength shift monitoring method, the sensitivity are changing with temperature as a quadratic function. The temperature sensitivity is about 0.04 nm/℃ at room temperature (25 ℃), which is even lower than the OSA resolution. Thus the temperature variations to the displacement sensing can be ignored.
7. Is it possible, by using the notch wavelength shift and the cross-correlation information to probe two parameters simultaneously (say displacement and temperature)?
Unfortunately, because the effects of the temperature and the displacement on the optical path difference cannot be separated, the corresponding temperature changes cannot be eliminated by the method of cross-correlation calculation. However, by using the polarization maintaining fiber to fabricate the F-P interferometer may demodulate the displacement change and the temperature change at the same time. Because the sensitivity weighting factor of the displacement and the temperature are different at two orthogonal polarization states.
8. Comparison with other sensors results is missing.
We have added the discussions about the displacement resolution comparisons from line No.181 to 194:
The displacement sensing can be converted to the strain sensing after calculation, the strain sensitivity comparison between the proposed F-P interferometer and other interferometer types sensors are shown as table.1.
Table 1. Comparison between references based on interferometer type structure and the sensor in this paper.
Sensor | Sensitivity | 3dB bandwidth | Ratio of bandwidth to sensitivity |
The proposed F-P | 6.4 pm/με | 0.15 nm | 23.4 με |
M–Z[1] | 1.19 pm/με | 4.3 nm | 3613 με |
SMS structure[2] | 42.5 pm/με | 7.3 nm | 171.8 με |
We can see that the sensitivity of the proposed F-P interferometer is higher than the M-Z structure while the 3dB bandwidth is much narrower. The sensitivity of the proposed F-P interferometer is less than that of the SMS structure, but the 3dB bandwidth of the proposed F-P interferometer is much less than that of the SMS structure. The ratio of the bandwidth to sensitivity is a very important parameters which is often used to represent the sensing resolution. The sensing resolution of the proposed F-P interferometer is higher than that of the two common interferometer type sensors.

Reviewer 2 Report
The authors propose a new displacement sensor based on a chirped Fabry-Perot interferometer with microfiber Bragg gratings. The authors showed the theoretical model and experimental results.
I recommend publication after revisions.
My comments are below:
1. The Introduction needs to be written more clearly. What did the authors want to achieve? Displacement sensing, small interferometer, optical spectrum chirping, or microwave signal generation?
2. The manuscript does not clearly state the principle of why one FBG acts as an FP interferometer.
I understand that the microfiber Bragg gratings inscribed on the two tapers act as two cavity mirrors, and are formed a resonator (interferometer).
3. Line No. 31: “Currently, fiber tip F-P interferometers can be fabricated by a variety of methods, such as focused ion beam micromachining, chemical etching, laser micro-machining and self-guiding photo polymerization.”
Is this topic about fabrication process of fiber tip F-P interferometers needed for the manuscript?
4. Line No. 34, 50, and 123: It is better to correct “characteristic” to “characteristics.”
5. Line No. 34: “Usually the microfiber 34 F-P interferometer is achieved by inscribing two mFBGs at the two taper regions of the microfiber.”
Why are not mFBGs integrated in the straight region?
6. Line No. 36: “monotonically decreasing FSR”
Why is such a characteristic required?
7. Line No. 54: “When the microfiber diameter reduces to only several micrometers, the microfiber can be treated as the “core” and the ambient air can be treated as the “cladding”, and the effective refractive index neff strongly depends on the microfiber diameter.”
I think that this explanation is difficult to understand for the readers because a fiber has “core” and “cladding” originally.
I understand that the refractive index sharply decreases because the light wave cannot be confined in the cladding and the evanescent wave in the air region increases.
8. Line No. 90: Indent value is too large.
9. Line No. 94: “tapper” must be corrected to “taper.”
10. Line No. 94: “We can guarantee that in the short wave region the interference conditions are satisfied, because the grating region at the minimum radius can be considered as both reflected gratings of the left taper and the right taper, and the cavity length is zero. While at the long wave region, when one grating area (could be the left taper or the right taper) is longer, the interference won’t exist.”
I cannot understand this theoretical principle.
11. Line No. 159: “microfiber Bragg grating” must be corrected to “microfiber.”
12. Line No. 192: Journal title “Express” of reference No. 10 must be corrected to “Opt. Express.”
Author Response
Reviewer comment
We thank the reviewer for the comments and recommendations.
1. The Introduction needs to be written more clearly. What did the authors want to achieve? Displacement sensing, small interferometer, optical spectrum chirping, or microwave signal generation?
We are sorry that our introduction is disorganized. We have rewritten the introduction. The manuscript proposed a new F-P interferometer sensor that can be potentially applied in high speed displacement sensing. The chirped spectra characteristics are the advantages of the proposed F-P interferometer for it can be used for the cross-correlation demodulation method to improve the sensing performance. The proposed F-P interferometer sensor also has potentials in the microwave signal generation. The rewritten introduction is as followed:
The displacement is an important structural mechanical parameter that used for damage identifications. The optical interferometer type of displacement sensor has the advantage of high sensitivity, small size and immunity to electromagnetic interference. The interferometer type displacement sensor usually has multi peaks and deeps. When demodulating the wavelength shift of a sensor with a uniform FSR, it is very difficult to mark one of the (notch) wavelengths as the sensing wavelength [1-3]. Because when the sensing wavelength shift is more than one FSR, it’s hard to trace the chosen sensing wavelength. But when the FSR is changing monotonically, the tracing of the chosen sensing wavelength becomes very easy [1,2]. Moreover, we can make cross-correlation process to the chirped spectra to achieve single peak. Marking one peak in the spectra is easier compared with the marking of the multi-peaks, especially in the high speed time domain sensing demodulation system [7, 8]. The spectral chirped optical fiber sensor can be also used in microwave photonics [9]. For example, in the field of microwave photonics radar system, the chirped microwave signals can be generated with optical spectral shaping techniques and frequency-to-time mapping techniques.
2. The manuscript does not clearly state the principle of why one FBG acts as an FP interferometer. I understand that the microfiber Bragg gratings inscribed on the two tapers act as two cavity mirrors, and are formed a resonator (interferometer).
We are sorry that our description for the principle has confused the reviewers. We have added the figure 3 and corresponding explanations. Although we only use one grating, there is actually interference. The long FBG can be separated into two gratings that inscribed on the two symmetrical regions around the microfiber, and their demarcation point is the minimum diameter of the microfiber. The corresponding explanations are from line No. 98 to 117:
Otherwise there will be no interference at the mismatched wavelength area at both the short wavelength region and the long wavelength region as shown in figure 3 (a).
Figure 3(a) The structure of the two separated FBGs based F-P interferometer and the reflected spectra of the FBG; (b) The structure of the one FBG based F-P interferometer and the reflected spectra of the FBG.
We can see that the two FBGs have the same length, but the left FBG (the red grating) is close to the zero point, while the right FBG is far from the zero point (the blue grating). Then the short wavelength of the left FBG spectra is smaller than that of the right FBG spectra, and the long wavelength of the left FBG spectra is smaller than that of the right FBG spectra. The overlapping region (the dashed area in the spectra) is where the interference occurs, and there is no interference region in the short wavelength range. Unlike the traditional fabrication process of the F-P interferometer with two discrete FBGs, one long FBG inscribed across the two taper regions is used in our experiment as shown in figure 3(b). We can see that the short wavelengths of the two FBGs are the same. At the zero point the length of the cavity is zero, which means the phase difference at the short wavelength is π (considering the half-wave loss). As the position is away from the zero point, the phase difference is increasing with wavelength, and the overlapping region (the dashed area) is the interference region. There will be extra spectra range for the right FBG which can be easily filtered out by additional optical filters.
3. Line No. 31: “Currently, fiber tip F-P interferometers can be fabricated by a variety of methods, such as focused ion beam micromachining, chemical etching, laser micro-machining and self-guiding photo polymerization.”
Is this topic about fabrication process of fiber tip F-P interferometers needed for the manuscript?
The manuscript proposed a new mode for the F-P interferometer sensor. We have deleted the description that is not related to the topic.
4. Line No. 34, 50, and 123: It is better to correct “characteristic” to “characteristics.”
Thank the reviewer for the careful review, we have corrected this spelling mistake in line No. 38, 54 and 141.
5. Line No. 34: “Usually the microfiber 34 F-P interferometer is achieved by inscribing two mFBGs at the two taper regions of the microfiber.”
Why are not mFBGs integrated in the straight region?
Theoretically speaking, there is no completely straight regions on the microfiber, especially when the length of the microfiber is relatively short. If the grating is written in the straight region, the spectra of the F-P interferometer will have a uniform FSR, and the chirped spectra characteristics cannot be achieved. The reason we need the chirped spectra characteristics is discussed from line No.22 to 29:
When demodulating the wavelength shift of a sensor with a uniform FSR, it is very difficult to mark one of the (notch) wavelengths as the sensing wavelength [1-3]. Because when the sensing wavelength shift is more than one FSR, it’s hard to trace the chosen sensing wavelength. But when the FSR is changing monotonically, the tracing of the chosen sensing wavelength becomes very easy [1,2]. Moreover, we can make cross-correlation process to the chirped spectra to achieve single peak. Marking one peak in the spectra is easier compared with the marking of multi-peaks, especially in the high speed time domain sensing demodulation system [7, 8].
6. Line No. 36: “monotonically decreasing FSR” Why is such a characteristic required?
We have added the description for the reason why we need a monotonically decreasing FSR in the introduction (line No. 22 to 29).
When demodulating the wavelength shift of a sensor with a uniform FSR, it is very difficult to mark one of the (notch) wavelengths as the sensing wavelength [1-3]. Because when the sensing wavelength shift is more than one FSR, it’s hard to trace the chosen sensing wavelength. But when the FSR is changing monotonically, the tracing of the chosen sensing wavelength becomes very easy [1,2]. Moreover, we can make cross-correlation process to the chirped spectra to achieve single peak. Marking one peak in the spectra is easier compared with the marking of multi-peaks, especially in the high speed time domain sensing demodulation system [7, 8].
7. Line No. 54: “When the microfiber diameter reduces to only several micrometers, the microfiber can be treated as the “core” and the ambient air can be treated as the “cladding”, and the effective refractive index neff strongly depends on the microfiber diameter.”
I think that this explanation is difficult to understand for the readers because a fiber has “core” and “cladding” originally. I understand that the refractive index sharply decreases because the light wave cannot be confined in the cladding and the evanescent wave in the air region increases.
We are sorry that we used a very imprecise description about the “core” and the “cladding”, we have removed this description. The situation when the microfiber diameter decrease to a few micrometres is just as the reviewer said, the light wave cannot be confined in the cladding and will transmit through the air region.
8. Line No. 90: Indent value is too large.
The indent value has been modified in Line No. 95 (has been merged into the previous paragraph). Thank the reviewer for the careful review.
9. Line No. 94: “tapper” must be corrected to “taper.”
Thank the reviewer for the careful review, we have corrected this spelling mistake in line No.111.
10. Line No. 94: “We can guarantee that in the short wave region the interference conditions are satisfied, because the grating region at the minimum radius can be considered as both reflected gratings of the left taper and the right taper, and the cavity length is zero. While at the long wave region, when one grating area (could be the left taper or the right taper) is longer, the interference won’t exist.”
I cannot understand this theoretical principle.
We have removed the inappropriate description and added figure 3 and corresponding explanation from line No. 98 to 117.
Otherwise there will be no interference at the mismatched wavelength area at both the short wavelength region and the long wavelength region as shown in figure 3 (a).
Figure 3. The F-P interferometer experimental reflected spectra under displacements of 1.05 mm (blue dot line) and 1.1 mm (red line).
We can see that when the two FBGs have the same length, but the left FBG (the red grating) is close to the zero point, while the right FBG is far from the zero point (the blue grating), then the minimum wavelength of left FBG spectra is smaller than the minimum wavelength of right FBG spectra, and the maximum wavelength of left FBG spectra is smaller than the maximum wavelength of right FBG spectra. The overlapping region (the dashed region) is the interference region, and we can see that there is no interference region in the short wavelength range. Unlike the traditional fabrication process of the F-P interferometer with two discrete FBG, one long FBG inscribed across the two taper regions is used in our experiment. We can see that the minimum wavelengths of the two FBGs are the same. In the zero point the length of the cavity is zero, which means the phase difference at the minimum wavelength is π (considering the half-wave loss). As the position is away from the zero point, the phase difference is increasing when the wavelength is increasing until the end of the left FBG, and the overlapping region (the dashed region) is the interference region. There will be extra spectra range for the right FBG which can be easily filtered out by additional optical filter.
11. Line No. 159: “microfiber Bragg grating” must be corrected to “microfiber.”
Thank the reviewer for the careful review, we have corrected this word mistake in line No. 205.
12. Line No. 192: Journal title “Express” of reference No. 10 must be corrected to “Opt. Express.”
Thank the reviewer for the careful review, we have corrected this spelling mistake in line No. 245.

Round 2
Reviewer 1 Report
The authors have performed the amendments as requested. I believe the manuscript can now be accepted.
Author Response
We thank the reviewer again for the valuable comments and recommendations